# Empowering Pre-Frail Older Adults: Assessing the Effects of a Community Nutrition Education Intervention on Nutritional Intake and Sarcopenia Markers

**DOI:** 10.3390/nu17091531

**Published:** 2025-04-30

**Authors:** Wei Leng Ng, Chung Yan Tong, Hiu Nam Chan, Theresa H. H. Kwek, Laura B. G. Tay

**Affiliations:** 1Department of Dietetics, Sengkang General Hospital, Singapore Health Services, Singapore 544886, Singapore; cherie.tong.c.y@skh.com.sg (C.Y.T.); candychanhn@gmail.com (H.N.C.);; 2Department of Geriatric Medicine, Sengkang General Hospital, Singapore 544886, Singapore; laura.tay.b.g@singhealth.com.sg; 3Geriatric Education and Research Institute, Singapore 768024, Singapore

**Keywords:** pre-frailty, nutrition status, community-dwelling, older adults, frailty, sarcopenia, protein intake, calcium intake

## Abstract

**Background/Objectives**: Early intervention combining nutrition optimisation with exercise can potentially prevent frailty progression and reverse pre-frailty in older adults. **Methods**: This 4-month study examined the effectiveness of nutrition education (without oral nutrition supplement use) as part of a multi-domain intervention on the nutritional status and intake of pre-frail community-dwelling older adults and its relationship with sarcopenia markers. **Results**: Amongst 172 participants (≥55 years), 5.8% were malnourished, with no significant change in nutritional status throughout the study. Post-intervention, participants consumed significantly higher daily calories, protein, protein per body weight (BW), and calcium (*p* < 0.001); protein intake at lunch (*p* = 0.001) and dinner (*p* = 0.004) also increased. However, 6-month post-intervention daily protein (*p* = 0.025), protein per BW (*p* = 0.039), and calcium (*p* = 0.015) decreased significantly. Sarcopenia markers (handgrip strength (HGS), five-time chair stand test (5STS), and short physical performance battery score (SPPB)) showed no significant difference post-intervention. Well-nourished participants had better HGS (*p* = 0.005), 5STS (*p* = 0.026), and SPPB (*p* = 0.039). Practical nutrition education effectively improved nutritional intake, but the effect was not sustained 6-months post-intervention. **Conclusions**: Optimising nutritional status with a focus on improving protein intake, especially at breakfast, to meet minimal intake to stimulate muscle protein synthesis can help prevent sarcopenia and frailty. Future studies should examine factors driving sustainable improvement to prevent frailty progression in this population.

## 1. Introduction

Singapore is facing a silver tsunami, with projections indicating that approximately one in four people will be aged 65 and above by 2030 [1]. This demographic shift presents significant challenges, as frailty is prevalent among older adults, with 5–30% of those aged 65 and above in Singapore experiencing frailty and pre-frailty [2,3]. Pre-frailty is an intermediate stage between being robust and frail, where individuals may exhibit one or two deficits in physical, psychological, or social domains, such as unintentional weight loss, weakness, or low physical activity [4]. It represents a critical intervention point, where early identification and management could potentially prevent its progression to frailty, thereby improving health outcomes and quality of life for older adults [5].

Nutrition plays a vital role in preventing the progression of sarcopenia and pre-frailty. Total daily protein intake and its consistent distribution across meals are vital for optimal muscle protein synthesis, retention, and strength [6]. Consuming 25–30 g protein or more per meal, or 0.4 g per kg body weight, or more high-quality protein has been associated with better muscle strength and lean mass retention in older adults [7,8,9,10]. Studies showed that higher protein intake in community-dwelling frail older adults led to improved muscle strength and performance such as handgrip strength and gait speed [11]. Calcium is also a critical nutrient for the maintenance of bone health and prevention of osteoporosis, which can exacerbate frailty. Oral nutritional supplements (ONS) are commonly prescribed for frailty and pre-frailty management in the community. While ONS were effective in improving nutritional and physical outcomes, their use may not be sustainable due to incurred cost and varying adherence [12,13].

Intervention programmes that integrated nutrition and physical activity demonstrated significant benefits in preventing frailty [14,15,16,17,18]. Systematic reviews emphasised that nutrition education programmes focusing on adequate protein and calcium intake can enhance physical function, mobility, and frailty outcomes [19]. These programmes, particularly when integrated with physical activity, yield measurable benefits, including improved muscle strength, better Short Physical Performance Battery scores, and enhanced overall nutritional status [19]. There is a paucity of local literature on the impact of multi-domain interventions targeting pre-frailty, apart from a recent local study [16]. In this study, the investigators utilised leucine-enriched protein supplementation combined with exercise, with reported improvements in the physical function and lean muscle mass of pre-frail older adults. A similar study conducted in Malaysia, the Frailty Intervention through Nutrition Education and Exercise (FINE) study, has yet to publish any results [20]. Therefore, this study aimed to investigate the effects of a nutrition education programme, as part of a multi-domain exercise and nutrition intervention, on nutritional status, intake, and sarcopenia markers amongst pre-frail community-dwelling older adults.

## 2. Materials and Methods

### 2.1. Study Design and Participants

Promoting Health LongevIty Through Mitigation and Prevention of Frailty in Community-dwelling Elderly (Pro-LIFE) is a 4-month pragmatic non-randomised controlled study of a multi domain exercise and nutrition intervention between June 2018 and December 2021. In this paper, we detail the immediate and sustained impact of the combined exercise and nutrition intervention on nutritional outcomes and sarcopenia parameters amongst the intervention group participants.

Potential participants were identified through our ongoing community frailty screening “Individual Physical Proficiency Test for Seniors (IPPT-S)”, which has been previously described [21]. Briefly, the screening platform was rotated around the void decks of public housing blocks, senior activity centres, and community clubs in the northeastern region of Singapore. Eligible participants (aged ≥55 years and can ambulate independently) in IPPT-S completed a multi-domain geriatric screen that included an assessment of mood (Geriatric Depression Scale, GDS) [22], cognition (modified version of Chinese Mini Mental State Examination, CMMSE) [23], nutrition (Mini Nutritional Assessment-Short Form, MNA-SF) [24], and functional performance (Barthel’s Index for activities of daily living (BADL) and Lawton and Brody’s instrumental ADL (iADL)) [25,26]. Frailty status was assessed using the FRAIL scale, a simple questionnaire comprising 5 components: Fatigue, Resistance, Ambulation, Illnesses, and Loss of weight. A value of 1 point was assigned for each component, with scores 0 indicating robust, 1–2 pre-frail, and 3–5 frail, respectively [27]. Participants who fulfilled operational criteria for prefrailty—defined as (i) FRAIL score of 1–2 or (ii) weak grip strength and/or slow gait speed based on Asian Working Group for Sarcopenia cutoffs [28] despite a FRAIL score of 0—were invited to participate in the Pro-LIFE programme. Participants assessed to be pre-frail but declined enrolment served as the controls. The study was approved by SingHealth Institutional Review Board (2018/2353) on 5 June 2018. Informed consent was obtained from all participants involved in this study. The study is registered with ClinicalTrials.gov (Identifier: NCT04656938).

### 2.2. Intervention

The study has been previously described [29] but herein we provide further details regarding the nutrition education intervention. Eligible pre-frail participants were invited to enrol in Pro-LIFE programme comprising (i) once-weekly group-based exercise classes lasting 1 h each session (total of 16 sessions) with individually prescribed home exercises for maintenance between sessions and (ii) group-based nutritional education (six sessions). Group size was maintained at 8–10 participants to ensure that each participant received adequate attention. Nutrition education intervention aimed to facilitate healthy eating habits that achieved adequate calories, protein, and calcium intake to avoid frailty progression without the use of oral nutrition supplements. A trained nutritionist conducted six nutrition sessions over a four-month intervention period: two sessions per month in the first two months and one session per month in the third and fourth months. Six topics were covered in the sessions: Eat Smart Age Well, Eat Adequate Protein for Strong Muscles, Calcium for your strong bone, Eat Well Age Well (introduction to My Healthy Eating Plate), How to Shop Smart, and A Healthy Tea Break. Each session lasted for 1.5 h, varying across different modes of nutrition education. There were didactic teachings focusing on key nutrients for frailty prevention (calorie, protein, and calcium), food-based games that allowed participants to apply knowledge learnt from lessons to practice, and a grocery shopping trip (with sponsored vouchers) teaching them how to shop for healthier food within a budget to meet key nutrients requirements. During the COVID-19 pandemic and circuit breaker lockdown, we pivoted to online delivery of the educational sessions. Compliance to Pro-LIFE programme refers to attendance of at least 4 out of 6 nutrition education sessions.

### 2.3. Outcome Measures

#### 2.3.1. Nutritional Status

Nutritional status was assessed by the trained nutritionist using the Subjective Global Assessment (SGA) 7-point measure. An overall nutritional status rating of well nourished (7 points), at risk of malnutrition (6 points), mildly malnourished (5 points), moderately malnourished (4 points), moderately to severely malnourished (3 points), severely malnourished (2 points), or very severely malnourished (1 point) was recorded for each participant. The SGA 7-point is a locally validated nutrition assessment tool [30], used in routine clinical practice. It is based on participant’s anthropometric, intake and gastrointestinal symptom history, together with physical examination of fat and muscle stores [31]. The SGA 7-point was conducted at four time-points: baseline (“Pre-intervention” immediately prior to commencement of intervention), post-intervention (at the end of the 4-month Pro-LIFE programme), 6-months post-intervention, and 12-months post-intervention after the Pro-LIFE programme.

The MNA-SF tool was also used to screen participants for their risk of malnutrition. It is a validated quick screening tool that is widely used in community settings, especially for community-dwelling older adults [24]. The MNA-SF consists of 6 items, with a total score ranging from 0 to 14. A score of 12–14 points indicates normal nutritional status, 8–11 points indicates at risk of malnutrition, and 0–7 points indicates malnutrition.

#### 2.3.2. Nutrition Knowledge

Participants were asked to complete a self-assessment of their knowledge on how to eat well to prevent or reverse frailty before the first and after the last education session. They were asked to rate their knowledge on a 4-point Likert scale: 1—very insufficient; 2—insufficient; 3—sufficient; and 4—very sufficient. A short quiz, comprising 5 to 6 bilingual (English, Mandarin) multiple-choice questions were created by study team dietitians and nutritionist to assess the participants’ nutritional knowledge. Participants had to complete the quiz pre- and post-nutrition education sessions 2, 3, and 4. The quizzes assessed specific knowledge domains: Session 2 focused on protein functions, sources and serving sizes; Session 3 evaluated understanding of calcium and vitamin D requirements and sources; and Session 4 tested comprehension of balanced meal planning using ‘My Healthy Plate’ and the identification of healthier food choices. Some quizzes included culturally relevant examples and local food items to ensure relevance for the Singapore population. Scores are presented in percentages, calculated based on the number of correct answers divided by the number of questions in the quiz. An increase in score from pre-to post-session was used to indicate improvement in nutritional knowledge.

#### 2.3.3. Dietary Intake

The nutritionist was trained to obtain a 3-day diet history (2 weekdays and 1 weekend) with the participants via face-to-face or over phone call. A protein and calcium checklist with the quantity and frequency of the food items consumed was also obtained. Participants’ diets were individually analysed using a nutrition analysis software (Nutritionist Pro™, Axxya Systems LLC, 2022, Redmond, WA, USA) to obtain the number of calories and amount of protein, protein per BW, and calcium consumed per day. Body weight used here refers to actual body weight for participants with BMI up to 24.9 kg/m^2^ and adjusted body weight for those with BMI 25 kg/m^2^ and above. Protein intake was further analysed over the six meals of the day—breakfast, morning tea, lunch, afternoon tea, dinner, and supper. These were measured at four time-points: baseline, post-intervention, 6-months post- and 12-months post-intervention.

#### 2.3.4. Sarcopenia Markers

Handgrip strength (HGS) was measured using a JAMAR hand dynamometer, with 2 trials for each hand and maximal value used for analysis. Lower limb strength and power was assessed using the time taken to complete 5 chair stands (5STS) [32]. We also scored each participant on the Short Physical Performance Battery (SPPB) [33] for a composite measure of physical performance, applying established cut-offs in the individual tests of gait speed, balance, and chair-stand. Using Asian Working Group for Sarcopenia (AWGS) 2019 cut-off values [28], low muscle strength was defined as HSG < 28 kg for males and <18 kg for females; impairment in physical performance was defined as 5STS ≥ 12 s with an SPPB score ≤ 9. These outcome measures were performed at three time-points: baseline (“Pre-intervention” at IPPTS), post-intervention, and 6-months post-intervention.

### 2.4. Statistical Analysis

Analysis was restricted only to intervention group participants as comprehensive nutritional outcome measures were not applicable to the control group. Available case analysis was utilised to maximise the use of all available data in view of small sample size. Patients identified with various degrees of malnutrition (SGA 1–5) and at risk of malnutrition (SGA 6) were categorised as ‘at risk/malnourished’ while those with an SGA 7 were classified as ‘well-nourished’. Data were analysed using SPSS (Statistical Package for the Social Science, Version 23, 2015, Armonk, NY, USA). Descriptive analyses were presented as absolute numbers and percentages, mean (SD) or median (IQR). The Chi-squared, *t*-test, and/or their non-parametric equivalent tests were used to compare and identify any significance between patients who had full datasets and those with missing data. The paired *t*-test was used to compare continuous variables between 2 time points of the study. Simple linear regression was utilised to find the correlation between protein intake and sarcopenia markers. A *p*-value < 0.05 was used to indicate statistical significance.

## 3. Results

There were 172 pre-frail participants recruited into Pro-LIFE. Table 1 shows the demographics of the participants. The majority were female (75%), Chinese (83%), within normal weight range (46%), and had a mean age of 71.2 years. Based on MNA-SF, 27.3% were at nutritional risk and 5.8% were diagnosed malnourished using the 7-point SGA. Before Pro-LIFE, most (58%) felt they had sufficient nutrition knowledge on knowing how to eat well to prevent or reverse frailty. They had a median HGS of 18.6 kg, a score of 12.0 for SPPB, and took a median time of 10.47 s to complete 5STS. Using the sarcopenia cut-offs presented by the AWGS 2019, the majority had low muscle strength (HGS 58%) but acceptable muscle performance based on their 5STS (67%) and SPPB scores (78%). In total, 59.3% of the participants were compliant to the Pro-LIFE programme and attended at least four education sessions. There was no difference in demographics, except for gender (*p* = 0.02), between those who were compliant and those who attended three or fewer lessons of the Pro-LIFE programme; more females were compliant to Pro-LIFE.

### 3.1. Nutritional Knowledge

Participants’ self-assessments of nutrition knowledge increased significantly post-intervention (pre: 2.5 vs. post: 3.1 score, *p* < 0.001), moving from insufficient to sufficient. The quiz scores for all sessions also improved significantly (*p* < 0.001) after the education sessions, with the highest increment after Session 2 on protein intake (Table 2).

### 3.2. Nutritional Status and Intake

The majority of participants maintained their nutritional status throughout the study period. There was no difference in nutritional status pre- and post-Pro-LIFE programme attendance (*p* = 0.167): pre- and 12 months post-Pro-LIFE (*p* = 0.054).

Participants consumed clinically and statistically significantly higher daily intakes of calories (*p* < 0.001), protein (*p* < 0.001), protein per BW per day (*p* < 0.001), and calcium (*p* < 0.001) post-Pro-LIFE programme completion as compared to baseline (Table 3). As shown in Table 4, intake significantly decreased 6 months post-intervention for daily protein (*p* = 0.025), protein per BW (*p* = 0.039), calcium (*p* = 0.015). Compared to baseline, there was no change found in dietary intake, except for calories (*p* = 0.004), at 12-month follow up (Table 3).

When comparing the intake of participants post-Pro-LIFE based on compliance, compliant participants consumed a higher amount of daily protein (*p* = 0.008), protein per BW (*p* = 0.008), and calcium (*p* = 0.042) post-intervention (Table 5).

For protein distribution across meals, lunch and dinner met the recommendations of 25–30 g protein per main meal. Protein intake at breakfast did not meet the recommended mealtime threshold, registering the lowest intake amongst the three meals across the day, and did not improve post-intervention (*p* = 0.848). Participants consumed 9% more protein at lunch (*p* = 0.001) and 11.5% more at dinner after attending the Pro-LIFE programme (*p* = 0.004) (Table 6). At the 12-month follow up, there was no difference in protein intake for all meals (Table 6). There was a 6% reduction (−0.85 ± SD 6.02 g) in protein intake at breakfast 12 months post-Pro-LIFE, albeit not statistically significant (*p* = 0.197).

### 3.3. Sarcopenia Markers

There was no significant difference in HGS (pre: 19.4 (±5.2) kg vs. post: 19.5 (±5.6) kg, *p* = 0.547), 5STS (11.23 (±4.40) s vs. 11.04(±4.50) s, *p* = 0.515), and SPPB (10.6 (±2.0) vs. 10.8 (±2.0), *p* = 0.295) post-intervention. We noted that well-nourished participants had better sarcopenia markers compared to at risk/malnourished participants at baseline—HGS (*p* = 0.005), 5STS (*p* = 0.026), and SPPB (*p* = 0.039) (Table 7).

Post-intervention daily protein intake per BW was found to correlate with 5STS timing (r = 0.260, *p* = 0.012) and SPPB score (r = 0.216, *p* = 0.037) but not HGS (Table 8). For every extra 1 g of protein per body weight consumed, participants will have better muscle performance—completing 5STS faster by 3.94 s and score 1.4 points higher for SPPB.

These correlations were not observed at the 6-month post-Pro-LIFE follow up. Sub-analysis based on AWGS 2019 in Table 9 and Table 10 revealed that participants above the 5STS cutoff (<12) consumed more protein/day (*p* = 0.021) and protein/kg/day (*p* = 0.044). Those above the SPPB cutoff (>9) also consumed more protein/kg/day (*p* = 0.040) (Table 10). Post-intervention significant results were not sustained at 6-months post-intervention.

## 4. Discussion

The Pro-LIFE programme demonstrated that a practical nutrition education programme without the use of ONS, as part of a multi-domain exercise and nutrition intervention, can improve nutritional knowledge and nutritional intake, despite the stability of the overall nutritional status of pre-frail community-dwelling older adults.

### 4.1. Dietary Intake

Participants achieved a significant increase in daily caloric, protein, protein per BW per day, and calcium intake in the immediate post-intervention period, especially those who were compliant to the Pro-LIFE nutrition education programme. Seino et al. (2017) supported the effectiveness of nutrition education in increasing the intake of calories and protein in pre-frail older adults, though in their study, there was no significant change in calcium intake of participants [34]. This may be related to the content design of the nutrition education programme. In our study, there was a special emphasis, and hence a dedicated session, on calcium intake for bone health that could have possibly positively influenced the calcium intake of our participants post-intervention.

### 4.2. Daily Protein Intake per BW

Multiple workgroups and papers have recommended a daily protein intake of 1.0–1.2 g/kg BW or even higher for older adults to prevent sarcopenia or for those with frailty [35,36]. The Singapore Clinical Practice Guidelines also suggested a minimum of 1.0 g/kg BW for sarcopenia prevention, emphasising the importance of resistance exercises and a protein-rich diet [37]. In this study, the participants’ mean baseline protein intake aligned with and had already exceeded the recommendations. Their protein intake increased further to 1.54 g/kg/day post-Pro-LIFE without the use of any ONS or protein supplementation. The effectiveness of protein supplementation remains controversial, with a systematic review including 62 randomised clinical trials showing limited benefit on functionality and only significantly reduced mortality risk for undernourished individuals [38]. The focus on providing nutrition education to improve nutritional knowledge as the main strategy to increase protein and calcium intake is thus an appropriate strategy for community-dwelling older adults who are not at high nutritional risk. The lack of sustained improvement in the post-intervention period suggests that continuing education or behaviour nudges may be necessary. However, it is notable that protein intake per BW was still well above recommended intake up to 12 months post-intervention. Given that the participants of this study were consuming sufficient protein intake per day during all time points of the study, it could provide some explanation to the non-significant difference in sarcopenia markers post-intervention and the lack of significant correlation between daily protein intake per BW and the 6-month post-Pro-LIFE sarcopenia measurements.

### 4.3. Protein Distribution Throughout the Day

Protein distribution across the day may be arguably as important as the total amount of protein consumed per day. In our study, participants consumed the lowest amount of protein at breakfast amongst the three main meals in the day, and this is not uncommon for community-dwelling older adults [39,40]. They did not achieve the minimal 20 g protein recommendations for muscle protein synthesis (MPS), even after receiving nutrition education focusing on protein intake. The intake of 25 g of protein per meal is recommended for older adults to maintain muscle mass and function, primarily due to the phenomenon of anabolic resistance, where ageing muscles require more protein to stimulate MPS effectively. Older adults experience anabolic resistance, necessitating higher protein intake to stimulate MPS effectively. Studies suggest that 25–30 g of high-quality protein per meal can maximise MPS in older adults, similar to younger individuals, but that older adults require a higher absolute protein dose due to this resistance [10,41]. For older adults with smaller body sizes, consuming 20–30 g of protein per meal is a practical approach to meet their protein needs without excessive intake. However, this can be challenging due to reduced appetite or dietary restrictions [41]. Furthermore, a German study by Bollwein et al. (2013) found that distribution, but not total amount of protein intake, was associated with frailty [40]. In his study, frail participants consumed significantly less protein at breakfast compared to pre-frail and robust participants, reinforcing the risk of frailty progression with a similar pattern of protein consumption amongst our prefrail participants. A further benefit was noted when a 10 g higher intake of protein at breakfast and lunch led to a significantly higher daily protein intake in community-dwelling older adults [42]. However, post-Pro-LIFE increase in total daily protein intake in our study could be attributed to the significant protein intake for lunch and dinner. This is evident by the fact that at the 12-month follow up post-Pro-LIFE, protein intake had returned to baseline for both lunch and dinner and there was no significant increase in total daily protein intake. Moving forward, nutrition education session should focus on encouraging older adults to increase protein intake in their daily breakfast, with the aim of encouraging even distribution of 20–30 g protein per main meal per day and not just the total amount of protein intake.

### 4.4. Sarcopenia Markers

There was no significant improvement in sarcopenia markers post-Pro-LIFE. However, we found significant correlations between daily protein per BW and muscle performance (5STS and SPPB), with parallel observation of higher daily protein per BW among participants whose 5STS and SPPB performance were above the AWGS 2019 cutoffs. This was not observed for HGS. The lack of significant improvement in sarcopenia markers post-Pro-LIFE programme completion may reflect the multifactorial nature of muscle function and physical performance and not just muscle mass, which is more likely impacted by nutritional intake. Factors such as individual adherence to the exercise regimens, baseline physical fitness, and the intervention’s duration may influence outcomes. Notably, our participants had relatively good baseline muscle performance, evident by the timing for 5STS (11.23 s) and SPPB score (10.6). The lack of correlation or significant difference in HGS despite exercise–nutrition intervention is not surprising given that HGS reflects static muscle strength rather than dynamic muscle performance, which may be more sensitive to changes in dietary protein and physical activity. This finding is consistent with a local study in which, after adjusting for confounding factors, Merchant and authors (2023) showed that combined nutrition and exercise intervention significantly improved 5STS timing and SPPB score in pre-frail older adults but the effect on HGS was lost [16]. An Austrian study using a home based exercise–nutrition intervention for pre-frail/frail older adults also noted that SPPB but not HGS was significantly better in the intervention group than the control group [43], though protein intake was not reported. Tieland et al. (2015) also showed that even after 24 weeks of whole-body resistance exercise training with protein supplementation, there was no measurable difference in HGS in pre-frail and frail older adults [44]. In our study, the exercises focused on strength, balance, and endurance training, and the predominant focus on lower rather than upper limb activities may have contributed to the lack of change in HGS. Our earlier study had also shown the weak correlation between 5STS performance with HGS, suggesting that these two measures are not equivalent strength test proxies of one another, with muscle mass more strongly related to HGS compared with a surrogate measure of lower body strength via 5STS [45]. Indeed, 5STS is influenced by strength, dynamic balance, and cardiorespiratory endurance, and thus is more representative of overall physical performance, reflecting the importance of muscle quality rather than mere muscle mass. HGS may be an easy, inexpensive, and non-invasive measurement but may not be an optimal method to assess the efficacy of exercise programme in pre-frail/frail older adults. However, we acknowledge that there are many studies that do show improvement in HGS in response to exercise intervention [46,47,48] in pre-frail/frail individuals and perhaps the difference lies in the nuances of the type of exercises prescribed and also the adaptation of different study populations, which is beyond the scope of discussion of this paper.

### 4.5. Nutritional Status

The nutritional status of participants plays a vital role in sarcopenia health. Our participants who were well nourished (SGA 7) had significantly better sarcopenia markers than those who were at risk/malnourished. Despite the focus on nutrition, nutritional status remained unchanged, with approximately one-quarter of participants, being assessed to be at risk of malnutrition or malnourished at follow-up. This could be due to us choosing to work with food choices and dietary changes rather than providing nutrition supplements as it is a more sustainable approach in the long run. The short-lived effectiveness of using supplements as an intervention has been shown in other studies. For example, in the local study by Merchant et al. [16], initial improvements in physical performance, fat-free mass, and appendicular skeletal muscle mass for pre-frail participants in the intervention group receiving exercise and a leucine-enriched protein supplement were not sustained at 3 months post-intervention. In another study, frail participants receiving protein supplements in the form of milk protein concentrate for 24 weeks demonstrated improved muscle performance but not skeletal muscle mass [49]. In both studies, the authors ensured that the participants’ baseline protein intake was at least 1 g per kg BW per day, which meant that participants needed to consume sufficient protein intake from the diet at baseline to gain potential benefits from the protein supplementation. Therefore, for pre-frail older adults, nutrition intervention involving both nutrition education and supplementation is likely necessary to at least prevent further deterioration in muscle function and performance. Individuals who are malnourished or at risk of malnutrition should receive intensive nutrition care by dietitians beyond the community setting to address oral intake and nutritional status for prevention of frailty and sarcopenia. An approach focusing on dietary changes without supplementation may be inadequate, especially for malnourished older adults. This is consistent with the findings from a Spanish study [18] and the Prefrail 80 study [17], in which a nutrition education session, using principles of the Mediterranean diet as part of a multifactorial intervention, failed to impact nutritional status. Worsening nutritional status over time was associated with progression to frailty [17].

### 4.6. Strengths and Limitations

The strength of our study was the use of different modes of nutrition education, such as didactic teaching, food-based games, and grocery shopping trips, without the use of ONS to emphasise dietary changes; this makes the learning process engaging and also promotes more realistic dietary improvement in real life. Furthermore, the long prospective 19-month study period also allows for the assessment of both the short-term and long-term effects of the intervention. This helps in understanding the sustainability of the benefits and identifying any potential decline over time. There are several limitations. We acknowledge the small sample size and an under-representation of male participants (25%) in our study. This is attributable to the fact that pre-frail participants were identified from a larger community-based frailty screening, wherein there was a predominance of female participants [50]. We relied on self-reporting 24 h dietary recalls for assessment of their dietary intake, which may be subjected to recall bias. The study was also unexpectedly disrupted by the COVID-19 pandemic, which resulted in (1) the change in modality of the nutrition education sessions, of which the immediate and long-term impact on dietary changes due to this change remains uncertain, and (2) loss to follow-up, resulting in variable sample sizes across time points. This may have introduced (1) selection bias, as participants who were more technologically savvy potentially had better participant engagement and comprehension, and (2) attrition bias, as the lockdown period resulted in some participants missing out on education session(s) before we pivoted to online delivery and physical measurements could not be carried out at one time point. There was also suboptimal compliance to the nutrition education programme at 59%, which may have contributed to the diminished and/or lack of intervention effect, especially on sarcopenia markers.

The lack of sustained significant improvement at 6-months or 12-months post-intervention follow-up highlights the challenge of maintaining dietary changes and their associated functional benefits over time. This decline was also noted in the local [16] and other studies [14,34] and underscores the need for prolonged interventions or follow-up support to reinforce dietary habits and maintain functional gains. There has been increasing interest in and potential for co-designing research study methodologies with patients and/or target populations as research partners to strengthen the impact of community interventions [51,52]. It may be a potential consideration in future study designs for our local setting. Given that the sustainability of dietary changes was not monitored after the 4-month intervention in our study, nor discussed in other studies, future studies warrant the investigation of factors that can drive long-term behavioural changes in pre-frail community-dwelling older adults to prevent the progression of frailty.

## 5. Conclusions

A practical nutrition education intervention without the use of oral nutrition supplement was effective in maintaining nutritional status, increasing nutritional knowledge, and improving the dietary intake (calories, protein and calcium) of pre-frail community-dwelling older adults. Optimising nutritional status with the focus on improving protein intake, especially at breakfast, can help to prevent sarcopenia and frailty. Despite these improvements, they were not sustained at the 6-month or 12-month follow-up, which underscores the vital need of continuous, sustained interventions to ensure lasting improvements in dietary habits and physical function. Future studies should focus on studying factors that drive sustainable improvement in pre-frail community-dwelling older adults to prevent the progression of frailty.

## Figures and Tables

**Table 1 nutrients-17-01531-t001:** Demographics of the Pro-LIFE participants (n = 172).

Variable	
Age (years ± SD)	71.2 (±7.25)
Gender	
Male	43 (25.0%)
Female	129 (75.0%)
Ethnicity	
Chinese	143 (83.1%)
Malay	19 (11.0%)
Indian	8 (4.7%)
Others	2 (1.2%)
Compliance	
Completely absent	17 (9.9%)
Attended 1–3 classes	53 (30.8%)
Attended at least 4 classes	102 (59.3%)
Body Mass Index (kg/m^2^) (n = 134)	24.4 (IQR 5.5)
Underweight	13 (9.7%)
Normal weight	62 (46.3%)
Overweight	38 (28.3%)
Obese	21 (15.7%)
Knowledge Questionnaire Pre-Pro-LIFE (n = 127)	3.0 (IQR 1.0)
Very Insufficient	9 (7.1%)
Insufficient	45 (35.4%)
Sufficient	64 (50.4%)
Very Sufficient	9 (7.1%)
MNA-SF (n = 168)	13 (IQR 3)
Malnourished	4 (2.3%)
At Risk of Malnutrition	43 (25.0%)
Normal Nutritional Status	125 (72.7%)
7-point SGA (n = 154)	
Mildly Malnourished	9 (5.8%)
At Risk of Malnutrition	29 (18.8%)
Well Nourished	116 (75.3%)
HGS (kg) (n = 151)	18.6 (IQR 6.8)
Low muscle strength?	
Yes	88 (58%)
No	63 (42%)
5STS (sec) (n = 150)	10.47 (IQR 4.62)
Low muscle performance?	
Yes	50 (33%)
No	100 (67%)
SPPB (score) (n = 151)	12.0 (IQR 2)
Low muscle performance?	
Yes	34 (22%)
No	117 (78%)

**Table 2 nutrients-17-01531-t002:** Pre- and post-quiz scores for Sessions 2, 3, and 4.

Quiz	Pre (±SD)	Post (±SD)	Change (±SD)	*p*
S2 ^a^	43.9% (28.6)	79.2% (26.6)	35.3% (29.1)	<0.001 *
S3 ^b^	65.8% (18.5)	83.0% (17.1)	17.2% (19.5)	<0.001 *
S4 ^c^	45.3% (28.8)	66.7% (23.5)	21.4% (26.8)	<0.001 *

^a^ n = 119; ^b^ n = 100; ^c^ n = 89. * statistically significant.

**Table 3 nutrients-17-01531-t003:** Comparison of nutritional intake pre- vs. post- and pre- vs. 12-month post-intervention.

Daily Intake	Pre	Post	*p*	Pre	12-Months Post	*p*
	Mean (±SD)			Mean (±SD)		
**Calories (kcal/day) ^a^**	1421.4 (258.3)	1560.3 (231.3)	<0.001 *	1413.7 (318.2)	1501.0 (269.9)	0.004 *
**Protein (g/day) ^b^**	75.08 (14.85)	83.30 (15.52)	<0.001 *	76.41 (16.77)	77.26 (17.01)	0.643
**Protein Per Body Weight (BW) (g/kg/day) ^c^**	1.40 (0.36)	1.54 (0.35)	<0.001 *	1.44 (0.37)	1.46 (0.36)	0.482
**Calcium (mg/day) ^d^**	781.8 (326.4)	939.3 (348.5)	<0.001 *	796.0 (335.1)	840.2 (345.2)	0.190

^a^ n = 102 for pre- vs. post-Pro-LIFE; n = 88 for pre- vs. 12 months post-Pro-LIFE. ^b^ n = 102 for pre- vs. post-Pro-LIFE; n = 88 for pre- vs. 12 months post-Pro-LIFE. ^c^ n = 102 for pre- vs. post-Pro-LIFE; n = 87 for pre- vs. 12 months post-Pro-LIFE. ^d^ n = 102 for pre- vs. post-Pro-LIFE; n = 88 for pre- vs. 12 months post-Pro-LIFE. * statistically significant.

**Table 4 nutrients-17-01531-t004:** Nutritional intake comparison post- vs. 6-months post-intervention.

Daily Intake	Post-Intervention	6-Months Post-Intervention	*p*
	Mean (±SD)		
**Calories (kcal/day)**	1556.4 (241.2)	1552.1 (242.0)	0.885
**Protein (g/day)**	84.47 (16.28)	80.66 (15.07)	0.025 *
**Protein Per BW (g/kg/day)**	1.56 (0.36)	1.50 (0.34)	0.039 *
**Calcium (mg/day)**	953.8 (358.9)	865.7 (316.3)	0.015 *

* statistically significant.

**Table 5 nutrients-17-01531-t005:** Nutritional intake post-Pro-LIFE between compliant and non-compliant participants.

Daily Intake	Non-Compliant (n = 23)	Compliant (n = 84)	*p*
**Calories (kcal/day)**	1506.2 (±245.0)	1566.5 (±230.6)	0.276
**Protein (g/day)**	75.28 (±14.32)	84.87 (±15.18)	0.008 *
**Protein Per BW (g/kg/day)**	1.36 (±0.32)	1.57 (±0.34)	0.008 *
**Calcium (mg/day)**	791.2 (±333.9)	958.5 (±348.9)	0.042 *

* statistically significant.

**Table 6 nutrients-17-01531-t006:** Protein distribution across the main meals in pre- vs. post- and pre- vs. 12-month post-intervention.

Daily Intake	Pre	Post	*p*	Pre	12-Months Post	*p*
	Mean (±SD)			Mean (±SD)		
**Breakfast ^a^**	13.44 (5.41)	13.55 (5.14)	0.848	13.58 (5.88)	12.73 (5.53)	0.197
**Lunch ^b^**	26.93 (6.79)	29.35 (7.39)	0.001 *	26.95 (6.55)	27.55 (6.94)	0.548
**Dinner ^c^**	28.31 (9.09)	31.57 (9.84)	0.004 *	29.28 (9.11)	30.45 (9.43)	0.286

^a^ n = 101 for pre- vs. post-Pro-LIFE; n = 84 for pre- vs. 12 months post-Pro-LIFE. ^b^ n = 102 for pre- vs. post-Pro-LIFE; n = 86 for pre- vs. 12 months post-Pro-LIFE. ^c^ n = 100 for pre- vs. post-Pro-LIFE; n = 86 for pre- vs. 12 months post-Pro-LIFE. * statistically significant.

**Table 7 nutrients-17-01531-t007:** Baseline sarcopenia markers based on nutritional status.

Sarcopenia Markers	Well Nourished (SGA 7)	At Risk/Malnourished (SGA 1–6)	*p*
**HGS (kg)**	20.2 (±5.6)	17.2 (±4.9)	0.005 *
**5STS (sec)**	10.62 (±3.41)	13.18 (±6.16)	0.026 *
**SPPB (score)**	10.8 (±1.9)	9.9 (±2.7)	0.039 *

* statistically significant.

**Table 8 nutrients-17-01531-t008:** Correlation between post-intervention daily protein intake per BW (g/kg/d) and post-intervention sarcopenia markers (n = 93).

Sarcopenia Markers	B	r	*p*
**HGS**	−1.25 (95% CI: −4.817–2.313)	0.073	0.487
**5STS**	−3.94 (95% CI: −6.988–−0.898)	0.260	0.012 *
**SPPB**	1.40 (95% CI: 0.083–2.717)	0.216	0.037 *

* statistically significant.

**Table 9 nutrients-17-01531-t009:** Post-intervention daily protein intake (g/d) based on AWGS 2019 cutoffs.

Sarcopenia Markers	Daily Protein Intake (Mean (±SD))		*p*
	Above AWGS Cutoff	Below AWGS Cutoff	
**HGS**	84.1 (16.2)	82.2 (15.5)	0.553
**5STS**	85.3 (15.7)	77.7 (14.7)	0.021 *
**SPPB**	84.6 (15.2)	76.7 (16.5)	0.073

* statistically significant.

**Table 10 nutrients-17-01531-t010:** Post-intervention daily protein intake per BW (g/kg/d) based on AWGS 2019 cutoffs.

Sarcopenia Markers	Daily Protein Intake Per BW (Mean (±SD))		*p*
	Above AWGS Cutoff	Below AWGS Cutoff	
**HGS**	1.47 (0.32)	1.55 (0.35)	0.553
**5STS**	1.56 (0.33)	1.41 (0.34)	0.044 *
**SPPB**	1.55 (0.33)	1.39 (0.36)	0.040 *

* statistically significant.

## Data Availability

The original contributions presented in this study are included in the article. Further inquiries can be directed to the corresponding author.

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
