# Peer review of "Empowering Pre-Frail Older Adults: Assessing the Effects of a Community Nutrition Education Intervention on Nutritional Intake and Sarcopenia Markers"

_nutrients, 2025, doi:10.3390/nu17091531_

Round 1
Reviewer 1 Report
Comments and Suggestions for Authors
Dear authors,
thank you for this study regarding nutritional education in a early frailty cohort of patients.
I have no immediate remarks, authors indicate clearly the results of the study. Unfortunately the effects of the intervention were small, and short lived. Singapore is ofcourse a very affluent state with resources that rival many Western Countries. I wonder how cost-effective these education sessions are, how happy the participants were with the content, how generalisable the results are to other countries, and how patient selection does influence results.
Reviewer 2 Report
Comments and Suggestions for Authors
Attached.

Reviewer 3 Report
Comments and Suggestions for Authors
This study examines the effects of a community-based nutrition education intervention on nutritional intake and sarcopenia-related outcomes in pre-frail older adults living independently. It explores how targeted education focusing on protein and calcium intake can modulate key health parameters such as muscle strength, physical performance, and nutritional status. The findings demonstrate that nutrition education significantly improves daily caloric, protein, and calcium intake immediately post-intervention, with partial maintenance over a 12-month follow-up. Notably, improvements in protein intake were associated with better physical performance measures such as the five-time chair stand test (5STS) and the Short Physical Performance Battery (SPPB), though static strength measures like handgrip strength remained unchanged. The study also highlights the critical influence of protein distribution across meals, particularly the persistent inadequacy of breakfast protein intake despite intervention efforts. Additionally, it considers the impact of compliance, session delivery methods, and pandemic-related disruptions on intervention outcomes. Utilizing a combination of subjective global assessments, dietary recalls, and physical performance tests, the research underscores the practical benefits and limitations of a food-based, non-supplemented strategy in preventing frailty progression. These results contribute to a growing body of evidence supporting early, sustainable dietary interventions as a cornerstone for maintaining musculoskeletal health and functional independence in aging populations.
However, I therefore have to point out some comments:
Abstract: "Improving protein intake, especially at breakfast" is highlighted but not sufficiently justified here. Add a short clause explaining why breakfast intake is critical.
Lines 53–54: Grammar mistake: “it may not sustainable”. Correct to “it may not be sustainable”.
Lines 76–77: The design is called "non-randomised controlled," but no further justification is given.
Lines 117–118: COVID-19 disruptions are briefly mentioned but not discussed in terms of potential bias introduced.
Lines 174–175: Only intervention participants were analyzed without stating how missing data were handled.
Lines 208–210: The quizzes are not described in sufficient detail.
Lines 217–224 (Table 3 and dietary intake analysis): You reported significant changes post-intervention, but did not mention whether these were clinically meaningful.
Lines 249–253: No effect on sarcopenia markers, but no power calculation for secondary outcomes is mentioned.
Lines 341–354: The finding that 5STS improved but not HGS needs deeper exploration.
Lines 373–384: Nutritional status did not change significantly; this major finding is underplayed. Discuss more critically whether dietary education alone is sufficient for malnourished/pre-frail groups without supplementation.
Comments on the Quality of English Languageno
Round 2
Reviewer 3 Report
Comments and Suggestions for Authors
no
Comments on the Quality of English Languageno